# Evaluation of a Hypersensitivity Inhibitor Containing a Novel Monomer That Induces Remineralization—A Case Series in Pediatric Patients

**DOI:** 10.3390/children8121189

**Published:** 2021-12-16

**Authors:** Manami Tadano, Aya Yamada, Yuriko Maruya, Ryoko Hino, Tomoaki Nakamura, Seira Hoshikawa, Satoshi Fukumoto, Kan Saito

**Affiliations:** 1Division of Pediatric Dentistry, Department of Oral Health and Development Sciences, Tohoku University Graduate School of Dentistry, Sendai 980-8575, Japan; manami.tadano.e5@tohoku.ac.jp (M.T.); yamada-a@dent.tohoku.ac.jp (A.Y.); yuriko.maruya.c1@tohoku.ac.jp (Y.M.); ryoko.m@dent.tohoku.ac.jp (R.H.); tomoaki.nakamura.d2@tohoku.ac.jp (T.N.); seira.hoshikawa.e5@tohoku.ac.jp (S.H.); fukumoto@dent.tohoku.ac.jp (S.F.); 2Section of Oral Medicine for Children, Division of Oral Health, Growth and Development, Faculty of Dental Science, Kyushu University, Fukuoka 812-8582, Japan

**Keywords:** amelogenesis imperfecta, enamel dysplasia, enamel hypoplasia, tooth sensitivity, molar incisor hypomineralization, C-MET, MDCP, recently erupted permanent tooth, pediatric dentistry

## Abstract

Background: Recently, tooth deformities have been frequently encountered by pediatric dentists. Severe enamel hypomineralization sometimes induces pain such as hyperesthesia, but composite resin restoration is difficult because it often detaches without any cavity preparation. Resin-based hypersensitivity inhibitors for tooth physically seal the dentinal tubules. It was reported that hypersensitivity inhibitor containing novel adhesive monomers forms apatite and induces remineralization in vitro. Therefore, these case series assessed the clinical effects of remineralization and the suppression of hypersensitivity by Bio Coat Ca (Sun Medical, Shiga, Japan). Methods: After mechanical tooth cleaning was performed, the hypersensitivity inhibitors were applied and cured by light exposure. Changes in hypersensitivity were determined by visual analog scale (VAS). The improvement of hypomineralization was evaluated by the change in color tone based on the digital images of intraoral photographs. Results: After repeated monthly treatments, these cases showed decreased hypersensitivity after the fourth application, while the opaque white and brownish color improved on the seventh application. Conclusion: This novel hypersensitivity inhibitor with calcium salt of 4-methacryloxyethyl trimellitic acid (C-MET) and 10-methacryloyloxydecyl dihydrogen calcium phosphate (MDCP) not only suppressed hypersensitivity but also improved cloudiness and brown spots in recently erupted permanent teeth in presented cases.

## 1. Introduction

Tooth deformities (amelogenesis imperfecta, enamel dysplasia, enamel hypoplasia, and hypomineralization) is a common disorder encountered by pediatric dentists. Permanent molar incisor hypomineralization (MIH), hypomineralized second primary molars (HSPM), and hypoplasia-associated severe early childhood caries (HAS-ECC) have recently been reported as systemic factors of amelogenesis imperfecta. The morbidity varies between countries, ranging from 2.8% to 36.5% [1,2,3]. Ameloblasts differentiate through secretory, mineralization and maturation stages to form enamel. If a disorder occurs during the secretory stage, the enamel defect is called hypoplasia. Enamel hypoplasia is a firm enamel condition in which the tooth has pits, grooves, and thin enamel [4]. In contrast, if it occurs during the mineralization or maturation stage, it is called hypomineralization. Hypomineralization appears as discolored patches of soft or decayed bumpy enamel that are sensitive and cause pain [4]. Hypoplasia and hypomineralization may occur due to effects during a mother’s pregnancy, the perinatal period, and during the first year after birth. Vitamin D deficiency due to decreased sun exposure, nutrition problems such as malnutrition and obesity, tobacco and alcohol consumption, complications during pregnancy such as diabetes and hypertension, neonatal complications, low birth weight, and medications during infancy may also have an effect; however, the true causes remain unclear [5,6,7,8]. Previous studies reported no significant differences in the site of occurrence between the upper and lower regions, between the right and left sides, or between men and women [9]. Hence, hypomineralization may occur in any person or tooth. There is a continuity between HSPM and MIH, and patients with HSPM are correlated with an approximately 20% risk of future MIH [10]. Therefore, in cases of primary tooth hypomineralization that are caused by systemic factors, pediatric dentists need to be careful about caring for future permanent teeth. On the one hand, local enamel hypoplasia and hypomineralization of the permanent teeth can be induced by trauma and caries of the primary teeth [11,12,13,14]. There is a correlation between the incidence of dental caries and dental trauma [15,16] and dental trauma has increased in recent years [17]. In addition, the incidence of tooth deformities is increasing [18,19] and many people experience traumatic dental injuries [20,21]. From this background, it has become necessary to use dental coating materials even in children. The enamel of tooth deformities is opaque white in mild cases and brown or with defects in more severe cases. Moreover, subjective symptoms such as dentin hypersensitivity can also occur. Deformities of the anterior teeth cause not only functional but also esthetic problems. Increased requests for treatment of discoloration in the anterior teeth of patients who expect to have a better quality of life are anticipated. While Icon^®^ (resin infiltration) has reportedly improved white spots, it does not affect brown spots such as those resulting from hypomineralization [22]. Composite resin restoration is sometimes performed for hypomineralization with enamel defects. However, composite restorations are more likely to detach without cavity preparation and hypomineralization is less effectively etched [23,24]. Thus, crowns may be used in teeth with severe and extensive hypomineralization. Since overtreatment has been avoided recently and tooth deformities are at high risk for caries [25,26], pediatric dentists must observe the children periodically and perform preventive measures to avoid deterioration.

Hypersensitivity is induced not only by hypomineralization but also by enamel attrition, abrasion, tooth chipping, cracked tooth syndrome, and periodontal problems such as exposure of root dentin. If hypersensitivity occurs, an inhibiting treatment is typically applied. The symptoms of hypersensitivity are improved by low-power diode lasers, sodium fluoride varnish, and occluding dentinal tubule agents [27,28]. Hydroxyapatite is a good hypersensitivity inhibitor material because it greatly occludes the dentinal tubule in vitro [29]. However, no agents have been reported to improve both color tone and hypersensitivity; therefore, the development of new agents is a pressing issue. Recently, Bio Coat Ca (Sun Medical, Shiga, Japan), a seal and coating material containing a Bioactive Monomer™ was released as a hypersensitivity inhibitor. Furthermore, because of its strong dentin adhesiveness, it can also be used as a light-cured dentin bonding system such as Hybrid Bond (Sun Medical) for the pretreatment of adhesive resin cement. The Bioactive Monomer™ includes a calcium salt of 4-methacryloxyethyl trimellitic acid (C-MET) and 10-methacryloyloxydecyl dihydrogen calcium phosphate (MDCP) (Figure 1). Tooth remineralization has been proposed to occur in vitro by apatite formation [30]. Furthermore, Bioactive Monomer™ inhibits biofilm formation by *Streptococcus mutans* [31]. It was also shown to act on odontoblasts and promote their differentiation [32]. This new monomer may clinically contribute not only to the suppression of hypersensitivity but also to remineralization and caries prevention, although it has not yet been reported. Consequently, Bio Coat Ca was repeatedly applied to recently erupted permanent teeth with discolored hypomineralization and hypersensitivity, and the clinical changes were investigated.

## 2. Materials and Methods

### 2.1. Ethical Application

Bio Coat Ca is approved in Japan as a sealant and coating (70860000), dentin adhesive (42483002), and hypersensitivity inhibitor (70926000). Written informed consent was obtained from the patients for the publication of this case report and the anonymous information included in the study. An ethical application at Tohoku University was approved as a case report (22740).

### 2.2. Desensitizer Treatment for Hypersensitive Teeth

A coating material that has been confirmed to induce calcification in vitro was applied to hypomineralization tooth with hypersensitivity and discoloration in children. The tooth surfaces were cleaned mechanically using a brush and non-fluoride toothpaste. BioCoat Ca is a hypersensitivity inhibitor and self-etch adhesive with Bioactive Monomer™ added to Hybrid Coat (Sun Medical). The monomer is composed of 4-methacryloxyethyl trimellitic anhydride (4-META), a brush containing a polymerization initiator, and calcium (Figure 2). One liquid drop was chemically reacted with a specific brush with a Bioactive Monomer™ by mixing it for 5 s. The tooth surface was then coated and maintained for 10 s. Air was blown for 5 s to thinly spread the liquid, and then cured by light irradiation. The non-polymerized layer on the tooth surface was wiped with an alcohol cotton ball. These treatments were performed once monthly.

### 2.3. Digital Image Analysis for Evaluation of Remineralization Ability

These intraoral photographs were taken at the pre-treatment before application and at the post-treatment after 7 applications. All intraoral images were taken using an EOS Kiss X5 camera (Canon, Tokyo, Japan) according to the commonly adopted method [33] (Figure 3a). Each photograph was normalized for color tone, contrast, size, and angle using Photoshop CS6 (Adobe, CA, USA). More than five images of teeth were taken from different angles. The most similar images were chosen using the superimposing function and the areas were normalized. The white balance was unified based on adjacent teeth and the color tone was regulated according to the gingival color. The photographs were analyzed to evaluate the remineralization ability of the agent. The fuzziness was set to 40 with color selection, three opaque white (Figure 3b) or brown areas (Figure 3c) were captured, and each color region was extracted. The extracted areas were measured using ImageJ (1.52a 23 April 2018) (http://imagej.nih.gov/ij/). Each measurement was performed in triplicate.

### 2.4. Subjective Assessment of Patients and Statistical Analysis

The changes in the colored areas were statistically evaluated. A two-tailed Student’s t-test was applied for the statistical analysis of two independent variables. All results were considered statistically significant if they had a *p*-value < 0.05.

### 2.5. Pain Evaluation of Hypersensitivity Using Visual Analogue Scale (VAS)

The pain of hypersensitivity was induced by cold water and blowing air. The pain of response was assessed by using a Numerical 0–10 VAS model [34]. The patients were asked to provide a numerical VAS rating with 0 indicating “no pain” and 10 indicating “intolerably severe pain”. When pointing between integers, it was evaluated as 0.5. The patient’s response was recorded before the application of hypersensitivity inhibitor, immediately after and every 1 month.

## 3. Case Presentation

### 3.1. Case 1: Hypersensitivity of the Permanent Teeth with Brown and Cloudiness Spots Caused by Primary Tooth Trauma

Case 1 was a female child aged 7 years and 11 months. She had visited the hospital with a chief complaint of cold water pain in the anterior mandible. She had a history of trauma to the anterior primary teeth, including the lower right central incisor, right lateral incisor, and left lateral incisor, at 3 years of age. Hypomineralized areas, brownish-white in color, were observed on the labial side of her lower bilateral central incisors (Figure 4). There was no past medical history. Genetic screening was not performed; the permanent tooth hypomineralization was thought to be caused by primary tooth trauma. The patient also complained of pain from air blowing and cold water, and the VAS value was 6.5. Immediately after the treatment to suppress the hypersensitivity, she no longer experienced pain with air or cold water, and VAS was zero. When patient came to the hospital one month later, her VAS score showed 4; therefore, the treatment was reapplied. After the fourth treatment, the hypersensitivity had not completely disappeared, and the VAS was 1. For the seventh treatment, patients’ VAS value of hypersensitivity pain were stable at 0.5–0. Furthermore, the surface of the brownish tooth had changed to appear almost cloudy after seventh treatment (Figure 5). During the process of this treatment, discolored devitalized teeth, gingival inflammation and percussion pain did not appear. Digital analysis indicated a pre-treatment cloudiness of 6331 pixels, which was significantly reduced to 65 pixels after treatment (Table 1). In addition, the area of brown color decreased by approximately six-fold, from 12,898 to 2118 pixels. These results suggest that both cloudiness and brown color disorder were significantly improved.

### 3.2. Case 2: Hypersensitivity of the Permanent Teeth with Cloudiness Spots Caused by Primary Tooth Trauma

Case 2 was a male child aged 8 years and 7 months. He visited the hospital with a chief complaint of pain in the anterior maxilla following exposure to cold water. He had a history of trauma to the anterior primary teeth at the age of one year, with composite resin repair of a fracture in the crown of the upper right primary central incisor. There was no past medical history. An abnormal position of the upper right permanent central incisor and clouding of the labial surface were observed, which were likely due to trauma to the primary teeth (Figure 6). Examination results indicated a VAS value of 6 for cold water and 7.5 for air blowing. Immediately after treatment, the patient no longer felt pain with air or cold water, VAS was zero. One month later, the VAS was 4 by cold water and 5 by air. The treatment was reapplied once monthly. The hypersensitivity had become acceptable to the patient and VAS was 2 after fourth treatment. During the seven treatments, the pain did not completely disappear, the VAS by cold water was 1–2, whereas the VAS by air was 2–4. While extensive clouding remained, the color tone was obscured and improved (Figure 7). During the process of this treatment, discolored devitalized teeth, gingival inflammation and percussion pain did not appear. Digital analysis showed significantly reduced cloudiness from 27,886 pixels to 7904 pixels (Table 2). The hypomineralized tooth was mostly cloudy, with a narrow area with a brown color. However, this area significantly decreased after treatment (*p <* 0.03). This result indicated that not only strong cloudiness but also slight brown color were significantly improved.

### 3.3. Case 3: Hypersensitivity of the Permanent Teeth with Brown Spots Caused by Primary Tooth Caries

Case 3 was a female child aged 8 years and 7 months. She had visited the hospital with a chief complaint of cold water pain in the left side of the maxilla. The left upper second primary molar was extracted because of apical periodontitis and root resorption due to severe caries, at 4 years of age. There was no past medical history. Dark brown hypomineralization was observed on the buccal tooth surface of the first premolars (Figure 8). The patient also complained of pain from air blowing and cold water, and the VAS value was 4. Immediately after the treatment to suppress the hypersensitivity, her VAS was zero. When patient came to the hospital one month later, her VAS score showed 1; therefore, the treatment was reapplied. During the seventh treatment, the hypersensitivity improved, and the VAS was 0. The dark brownish tooth surface of the first premolars was changed to pale brown (Figure 9). Digital analysis significantly reduced the area of the brownish tint from 4858 to 1755 (Table 3). On the other hand, cloudiness was not detected.

### 3.4. Case 4: Hypersensitivity of the Permanent Teeth with Brown and Cloudiness Spots

Case 4 was a male child aged 5 years and 9 months. He had visited the hospital with a chief complaint of cold water pain in the anterior mandible. There was no history of trauma and caries in the primary teeth and no other systemic history. The cause of hypomineralization in the permanent teeth was not determined. The brownish-white in color were observed on the labial side of his lower central incisors (Figure 10). The patient also complained of pain from air blowing and cold water, and the VAS value was 3. Immediately after the treatment to suppress the hypersensitivity, his VAS was zero. After one month, VAS was reduced to 0.5 and VAS was zero after four treatments. Seven treatments improved the color of the hypomineralization (Figure 11). Cloudiness areas improved from 6872 to 1903, and brown areas decreased significantly from 6595 to 1667 (Table 4).

### 3.5. Changes in VAS of Hyperesthesia and Remineralization in These Four Cases

The hyperesthesia inhibitor was applied once a month and continued for 7 months. The mean VAS of the pre-treatment was 5.25 ± 2.1 (Table 5). After one month, the mean VAS decreased to 2.50 ± 2.0, but the difference was not significant (*p* = 0.1). However, the mean VAS was reduced to 0.75 after four treatments, and pain of hypersensitivity was clearly suppressed (*p* < 0.02). Depending on the degree of hypomineralization, the hypersensitivity improved and there was little change after 4 months of repeated treatment since the mean VAS for 4 to 7 months was 0.75. The rate of remineralization ability was calculated from the area change of cloudiness and brown (Table 6). The improvement rate after 7 treatments was 80.9 ± 15.6% for cloudiness and 72.9 ± 26.2% for brownish color.

## 4. Discussion

Children aged 1–5 years are more exposed to primary tooth trauma due to immature walking abilities and reflexes [35,36,37,38]. Trauma to the primary teeth often results in abnormal eruption, hypoplasia, or hypomineralization of the succeeding permanent teeth [38,39,40]. Fluoride varnish was previously used for the treatment of hypomineralization and hypersensitivity; however, the degree of elution varied depending on the product and there was no clear evidence of its efficiency. New hypersensitivity inhibitors using resins, potassium nitrate, and glutaraldehyde have been developed to occlude dentinal tubule agents to treat hypersensitivity. Thus, with the evolution of adhesive restorative materials, the concept of minimal intervention is spreading among dentists. Carboxylic acid-based monomers such as 4-methacryloxyethyl trimellitic acid (4-MET) are often used as acidic monomers in self-etching primers or one-step bonds because they increase resin bonding to enamel and dentin, given the hydrophilic group’s potential to form chemical bonds. Recently, a split-mouth randomized double-blind clinical trial comparing fluoride varnish and bonding resin for hypersensitivity was reported [41]. Dentin hypersensitivity was improved in both groups. However, VAS improvement rates and long-term effects were more effective with resin. Therefore, the resin system should have high potential as a hyperesthesia inhibitor. The 4-MET is not only highly adhesive but also additional multifunctional monomers with anti-bacterial, durable and ion sustained release properties [42,43,44,45,46]. However, some dental adhesive materials such as monomers are also cytotoxic to odontoblasts and dental pulps [47,48,49,50]. Icon^®^ is an agent that uses a resin infiltration system to improve white spots. However, it is toxic to dental pulp [51] and does not improve hypersensitivity or brown spots. BioCoat Ca is a novel self-etch adhesive resin material that inhibits hypersensitivity. It shows high adhesive strength, inhibits biofilm formation, allows remineralization, and also has low cytotoxicity in vitro [30,31,32]. Therefore, this agent that is safe and caries-preventive may be suitable for children.

In this case series, self-etch adhesive containing C-MET and MDCP to form apatite was applied to the hypomineralization areas with hypersensitivity to evaluate the hypersensitivity suppression and remineralization effects. After 4 month, the mean VAS decreased from 5.25 to 0.75, and hypersensitivity was clearly suppressed. However, no significant improvement was observed after repeated application. The monthly treatment may need to be repeated four times to improve hypersensitivity. Until now, the presence or absence of hyperesthesia and the need to prevent caries have been one of the parameters used to determine the use of coating material or sealant. Since the material used this time contains a remineralized monomer, it was expected to promote calcification of hypomineralized enamel. This new coating will add hypomineralization with abnormal tooth color to the conventional parameters. In this study, these cases of hypomineralization teeth with hypersensitivity were recently erupted permanent teeth. Recently erupted teeth are insufficiently mineralized immediately after eruption; however, the calcium and phosphorus contained in saliva promotes calcification and maturation over several years [52]. Therefore, the treatment of recently erupted permanent teeth with remineralization-inducing agents such as C-MET and MDCP may be more effective. Recently, it has been reported that C-MET activates odontoblasts and induces mineralization [32]. Thus, the agent may not only have covered the dentinal surface, but may also activate odontoblasts through the dentinal tubules in the recently erupted permanent teeth, resulting in the formation of tertiary dentin. This result suggests that the agent induced remineralization of the tooth surface by forming apatite. The structure of primary and permanent teeth differ, in that primary teeth have a wider pulp cavity, thinner dentin and enamel, and lower calcification [53,54]. Since the characterization of primary teeth is similar to hypomineralization, the application of C-MET and MDCP might increase calcification and strengthen the teeth.

In recent years, research has focused on remineralizing agents [55]. As non-collagenous proteins (NCPs) are important for dentin remineralization, NCP analogues such as polyacrylic acid (PAA), polyvinylphosphonic acid (PVPA), and sodium trimetaphosphate (STMP) have been studied. For example, NCPs with high affinities for calcium ions and collagen fibrils regulate the nuclei and growth of minerals in biominerals, such as dentin phosphophoryn and dentine matrix proteins with highly phosphorylated serine and threonine residues. PAA stabilizes the amorphous calcium phosphate and converts it to apatite before it enters the dentin collagen fibrils. Furthermore, polyphosphate-containing biomimetic analogs such as PVPA or STMP are used to bind to the dentin collagen matrix and further attract the amorphous calcium phosphate nano-precursors into the collagen matrix. However, these are applied as pretreatment agents with NaOCl, EDTA, and phosphoric acid to completely expose the type I collagen in dentin. Furthermore, while phosphoric acid may cause the denaturation of type I collagen, it also requires a complicated process. The resin infiltration system requires pretreatment, including strong etching such as hydrochloric, and a long working time [56]. In contrast, BioCoat Ca did not require intricate pretreatment, and the treatment is completed in a short time. C-MET and MDCP may also be an effective agent for inducing tooth remineralization.

White spots are a major problem not only in pediatric and cosmetic dentistry but also in orthodontics, as the plaque that adheres around the bracket forms a white spot on the tooth surface. While demineralization prevention using fluoride may be considered, there is no clear evidence of its effectiveness [57]. Furthermore, acidulated phosphate fluoride reduces Ni-Ti strength [58]. Therefore, new approaches are required. BioCoat Ca is not only a hypersensitivity inhibitor but also a self-etch adhesive. It is recommended by the manufacturer as a pretreatment agent for adhesive resin cement such as Super-Bond (Sun Medical). If it is applied as a dentinal adhesive before not only orthodontic brackets but also inlays, crowns or bridges are cemented, it may increase strength and prevent demineralization. These new adhesive monomers may be applied in various fields, such as improving oral health, tooth appearance, and patient quality of life. However, an increased number of cases and further evidence are needed by applying this method to both permanent and primary teeth. Moreover, treated primary tooth will result in chemical compositional and structural analysis after shedding.

### Limitations of the Study

This study is a case series of a novel monomer that induces hyperesthesia suppression and remineralization. Currently, this product is only released in Japan and is not available worldwide, so the dentists who can use it are limited. Individual difference effects may occur because these four limited cases are the treatment of recently erupted permanent teeth. In this case series, the VAS was applied to children, so the values were widely. The hypomineralization and hypersensitivity improved but did not disappear completely with repeated treatments. The treatment was discontinued because no improvement was observed after more than seven treatments and was considered as a clinical endpoint. Since it was applied to permanent teeth, it was not possible to perform chemical composition and structural analysis.

## 5. Conclusions

This novel hypersensitivity inhibitor with C-MET and MDCP suppressed hyperesthesia in recently erupted permanent teeth in presented cases. The slight recurrence was observed after one month. Pain was significantly reduced and stabilized after four applications. The hypomineralization of brown color and cloudiness was improved by seven applications. However, this report is based on a limited sample size, further evidence will be provided in an increasing number of homogenous cases in the future.

## Figures and Tables

**Figure 1 children-08-01189-f001:**
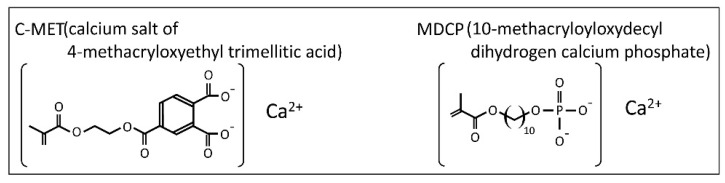
Chemical formulas of new adhesive monomers C-MET and MDCP.

**Figure 2 children-08-01189-f002:**
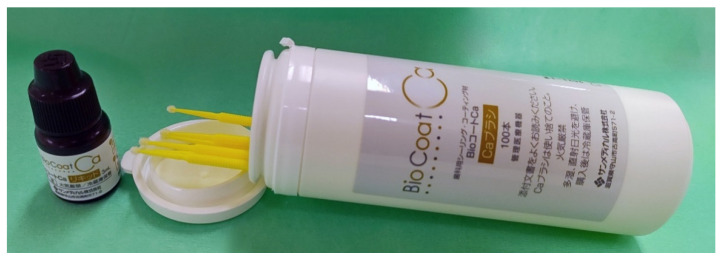
BioCoat Ca is approved as a hypersensitivity inhibitor and self-etch adhesive with Bioactive Monomer™ added to Hybrid Coat (Sun Medical). Bioactive Monomer™ consists of a liquid with 4-methacryloxyethyl trimellitic anhydride (4-META), a brush with a polymerization initiator, and calcium.

**Figure 3 children-08-01189-f003:**
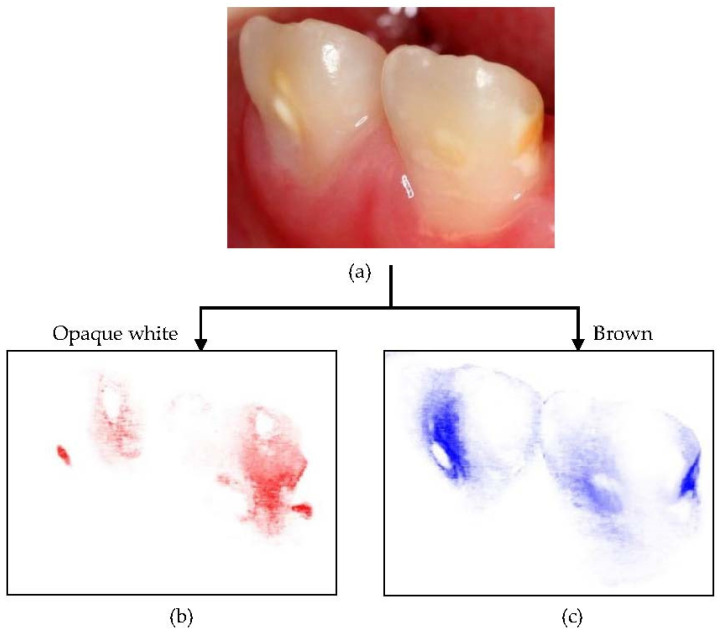
Opaque white (**b**; red) and brown (**c**; blue) areas were extracted from the enamel hypoplastic areas and photographed with a digital camera (**a**) using Photoshop.

**Figure 4 children-08-01189-f004:**
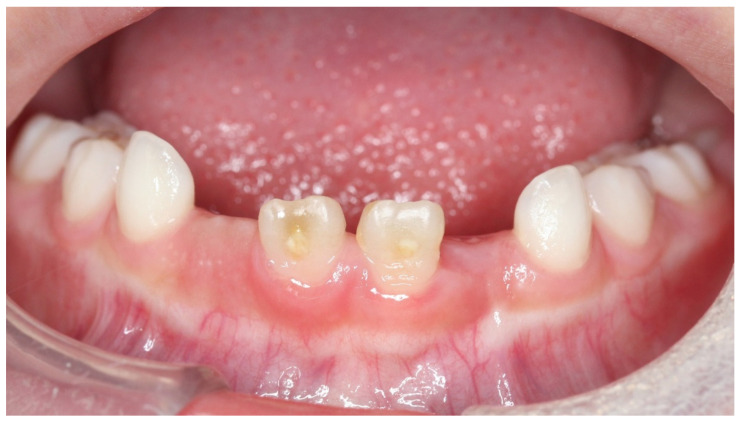
Case 1: Intraoral photograph before treatment.

**Figure 5 children-08-01189-f005:**
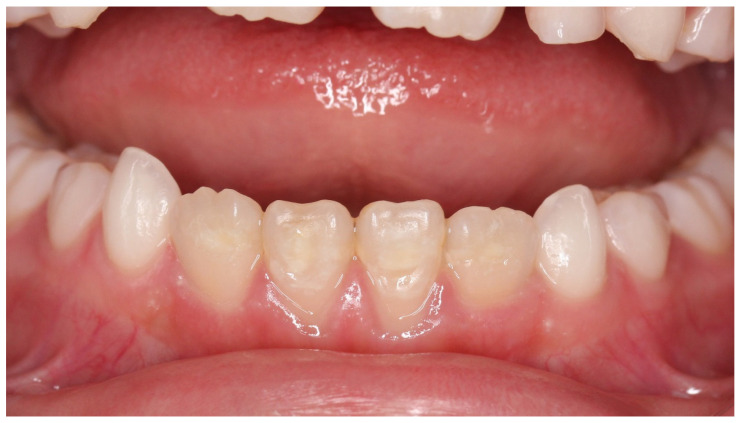
Case 1: Intraoral photograph after seven-times treatment.

**Figure 6 children-08-01189-f006:**
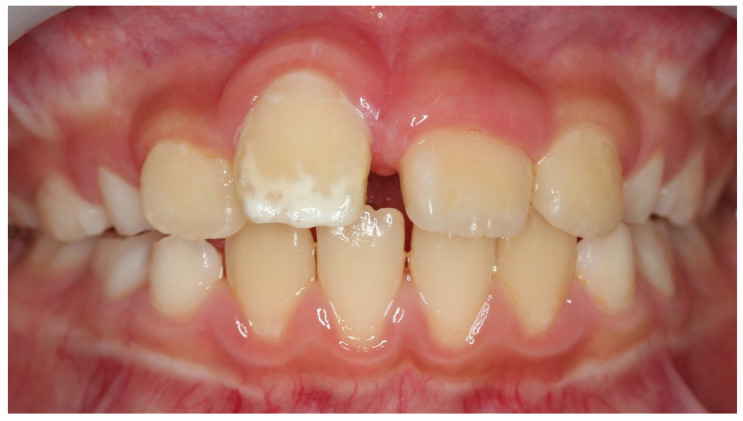
Case 2: Intraoral photograph before treatment.

**Figure 7 children-08-01189-f007:**
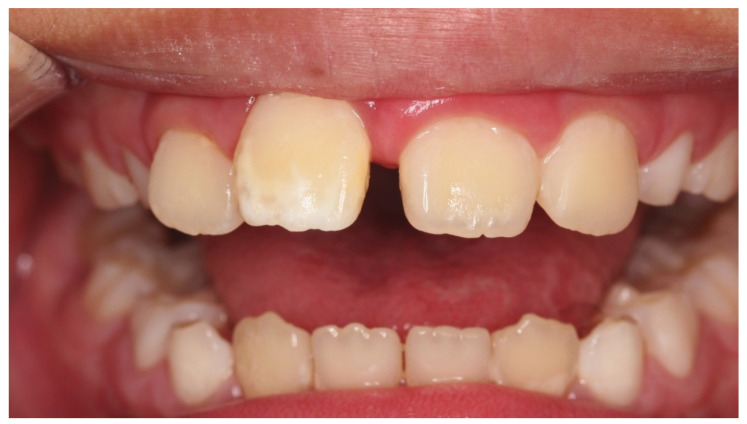
Case 2: Intraoral photograph after seven-times treatment.

**Figure 8 children-08-01189-f008:**
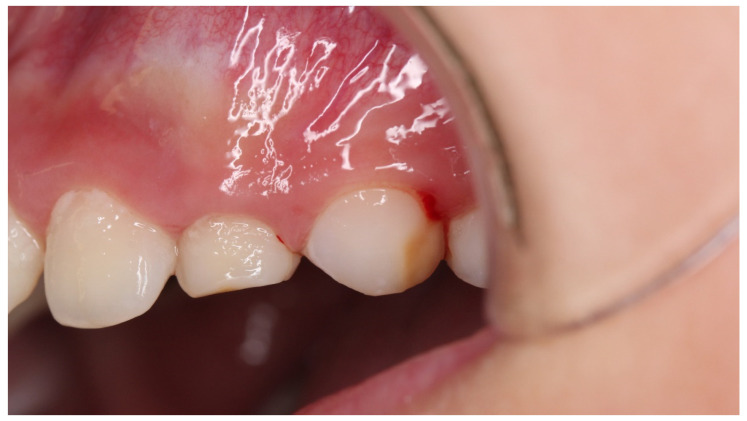
Case 3: intraoral photograph before treatment.

**Figure 9 children-08-01189-f009:**
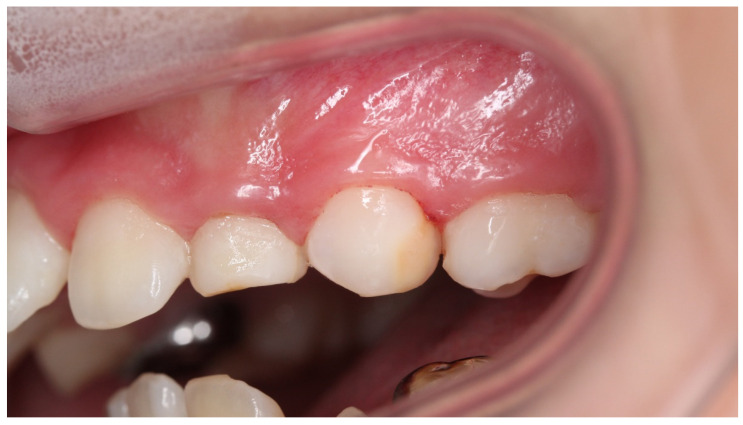
Case 3: Intraoral photograph after seven-times treatment.

**Figure 10 children-08-01189-f010:**
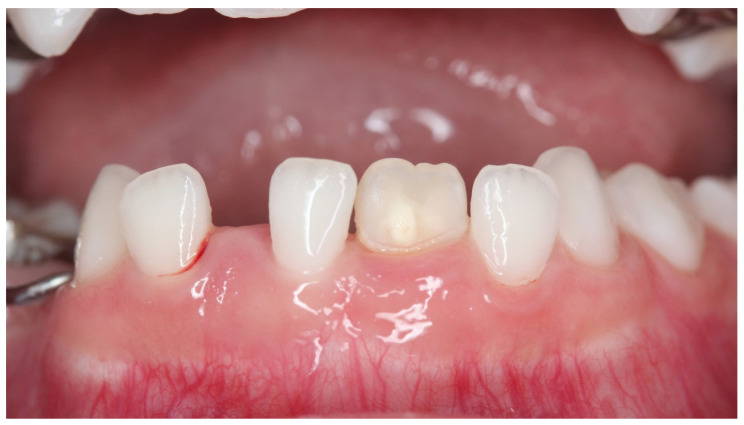
Case 4: Intraoral photograph before treatment.

**Figure 11 children-08-01189-f011:**
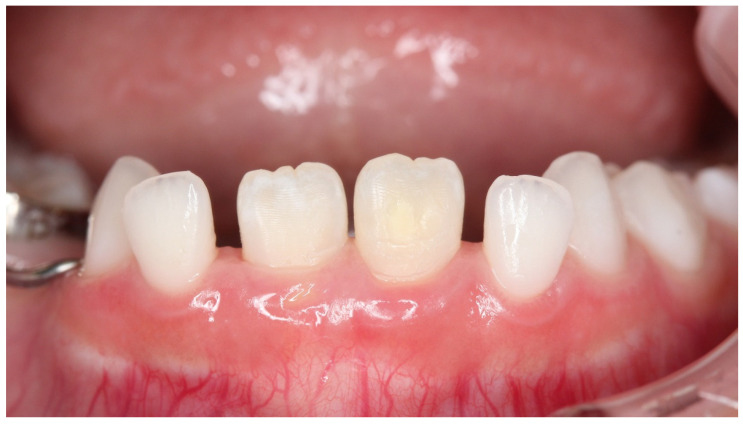
Case 4: Intraoral photograph after seven-times treatment.

**Table 1 children-08-01189-t001:** In Case 1, the cloudiness and brown areas from the pre-treatment (before applications) and post-treatment (after 7 applications) images were measured as pixels by digital image analysis. The analysis was performed three times and the mean measures and standard deviations were calculated (mean ± SD). The significance of the differences in areas before and after treatment was evaluated with *p*-values by *t*-test.

Case 1	Pre-Treatment (Pixels)	Post-Treatment (Pixels)	*p*-Value
Cloudiness	6331 ± 1091	65 ± 48	*p <* 0.0005
Brown	12,898 ± 1223	2118 ± 923	*p <* 0.0003

**Table 2 children-08-01189-t002:** In Case 2, the cloudiness and brown areas from the pre-treatment (before applications) and post-treatment (after seven applications) images were measured as pixels by digital image analysis. The analysis was performed three times, and the mean and standard deviations of the measurements were calculated (mean ± SD). The significance of the differences in areas before and after treatment was evaluated with *p*-values by *t*-test.

Case 2	Pre-Treatment (Pixels)	Post-Treatment (Pixels)	*p*-Value
Cloudiness	27,886 ± 2341	7904 ± 1304	*p <* 0.0003
Brown	4541 ± 2040	122 ± 156	*p <* 0.03

**Table 3 children-08-01189-t003:** In Case 3, the cloudiness and brown areas from the pre-treatment (before applications) and post-treatment (after seven applications) images were measured as pixels by digital image analysis. The analysis was performed three times and the mean measures and standard deviations were calculated (mean ± SD). Cloudiness was not detected (N.D.). The significance of the differences in areas before and after treatment was evaluated with *p*-values by *t*-test.

Case 3	Pre-Treatment (Pixels)	Post-Treatment (Pixels)	*p*-Value
Cloudiness	N.D.	N.D.	
Brown	4858 ± 339	1755 ± 111	*p <* 0.0002

N.D. = Not detected.

**Table 4 children-08-01189-t004:** In Case 4, the cloudiness and brown areas from the pre-treatment (before applications) and post-treatment (after seven applications) images were measured as pixels by digital image analysis. The analysis was performed three times, and the mean and standard deviations of the measurements were calculated (mean ± SD). The significance of the differences in areas before and after treatment was evaluated with *p*-values by *t*-test.

Case 4	Pre-Treatment (Pixels)	Post-Treatment (Pixels)	*p*-Value
Cloudiness	6872 ± 442	1903 ± 580	*p <* 0.0003
Brown	6595 ± 102	1667 ± 671	*p <* 0.0003

**Table 5 children-08-01189-t005:** Hyperesthesia suppressing treatment were performed once monthly. The mean and standard deviations of the VAS were calculated (mean ± SD). The significance of the differences between pre-treatment and after treatment (1, 4 and 7 months) was evaluated with *p*-values by *t*-test.

	Pre-Treatment	1 Month	4 Months	7 Months
Case 1	6.5	4	1	0.5
Case 2 Case 3 Case 4	7.5 4 3	4.5 1 0.5	2 0 0	2.5 0 0
Mean ± SD	5.25 ± 2.1	2.50 ± 2.0	0.75 ± 0.95	0.75 ± 1.19
*p*-value		*p* = 0.1	*p* < 0.02	*p* < 0.03

**Table 6 children-08-01189-t006:** Hyperesthesia suppressing treatment were performed once monthly. The mean and standard deviations of the improvement rate of cloudiness and brown hypomineralization were calculated (mean ± SD). N.D. = not detected.

	Cloudiness	Brown
Case 1	98.9%	83.6%
Case 2 Case 3 Case 4	71.7% N.D. 72.3%	97.3% 36.1% 74.7%
mean ± SD	81.0 ± 15.6%	72.9 ± 26.2%

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
