# Peer review of "Evaluation of a Hypersensitivity Inhibitor Containing a Novel Monomer That Induces Remineralization—A Case Series in Pediatric Patients"

_children, 2021, doi:10.3390/children8121189_

Round 1

Reviewer 1 Report

Firstly, it should be stated that the article is well-written, especially in the introduction and discussion section. The use of newly-designed C-MET and MDCP adhesive monomers in pediatric patients should be praised for introducing an innovation. I can understand the desire to be an innovator - "the first" who puts something into use. However, in the case of such a trivial procedure in possibly such a large group of patients as children suffering from MIH, the case report form cannot be justified. The results of this research remains too subjective and do not meet the conditions of scientific soundness. This two cases slightly differ from each other, the clinical conditions are different and so the response of patients towards this desensitizer. In my opinion, with such a large number of reports on hypersensitivity and its treatment, the only acceptable form of such article is a RCT or case-control study on a homogeneous group of at least several dozen patients with similar MIH severity in both groups in order to be considered for publication in journal with around 2,9 IF score. In the present form, the results of the article are not binding in any way, and their possible publication may, in my opinion, give the impression that the use of the new substance in patients suffering from MIH is fully compliant with all the conditions set out by evidence-based medicine. I would like to encourage the authors not to abandon the intention to publish on the effects of using the innovative desensitizer and to extend the study to a larger number of patients with the preservation of activities such as randomization, double-blinding etc..

Author Response

Response to reviewer:

I appreciate your valuable comments. As you suggested, I also think it is necessary to write observational studies for firm evidence. So we are now increasing the number of cases in collaboration with other universities. However, as this agent is only available in Japan, it will take time to establish the evidence. This case report is the first clinical trial article using C-MET and MDCP. Based on this report, it may be sold worldwide and the number of users may increase. If so, many papers will be published and the accumulation of evidence will be accelerated. Then, I believe this article will be cited. However, as you pointed out, there are only two cases, so we added new cases and statistics for increased scientific soundness.

Reviewer 2 Report

Dear Authors,

Congratulations for your original and very interesting work. 

The topic is really interesting and novel and can truly add new inspiration to current literature. The work is very well presented and structured.

However, I have some minor concerns that I would like to be addressed. You will find some notes in the highlighted parts of the manuscript attached plus few comments/questions below. 

  • I enjoyed the introduction and materials and methods sections, very well structured. There is a 'Limitations of the study' section that is really useful and interesting. But I would put it as a subparagraph of the 'Discussion' and instead adding a CONCLUSION paragraph; it is strongly recommended.
  • Furthermore, it is true that it is a new material and only approved for use in Japan, but having only two clinical cases and only with incisors supporting your results, is a strong limitation. I would state that in the text and add in the conclusion paragraph suggestions for future researches.

Keep up the good work and congratulations!

Author Response

  • I enjoyed the introduction and materials and methods sections, very well structured. There is a 'Limitations of the study' section that is really useful and interesting. But I would put it as a subparagraph of the 'Discussion' and instead adding a CONCLUSION paragraph; it is strongly recommended.

Response to reviewer:

Thank you for your constructive comments. I put 'Limitations of the study' as a subparagraph of the 'Discussion' and instead adding a CONCLUSION paragraph.

  • Furthermore, it is true that it is a new material and only approved for use in Japan, but having only two clinical cases and only with incisors supporting your results, is a strong limitation. I would state that in the text and add in the conclusion paragraph suggestions for future researches.

Response to reviewer:

We increased the number of cases for further evidence. In addition, future research is described in the "conclusion" paragraph. We also recognize that there is strong limitation. It was described in the "Limitations of the Study" and "conclusion" paragraphs.

Reviewer 3 Report

This manuscript is interesting, but it is just two cases and it is not enough for s getting any conclusion. 

I suggest that authors present this manuscript  as the case report, or expand the research on at least 10 or 15 cases. Also, I suggest split mouth study design. 

Author Response

Response to reviewer:

Thank you for reviewer's comment. As you suggested, I also think it is necessary to write research article for firm evidence. Now we are increasing the number of cases and preparing articles. However, as this agent is only available in Japan, it will take time to establish the evidence. This is the first article to evaluate C-MET and MDCP in a clinical trial. Therefore, the manuscript was submitted in the Case Report Type.

Based on this report, I hope it may be sold worldwide and the accumulation of evidence will be accelerated soon. However, as you pointed out, there are only two cases, so we added new cases and statistics for increased scientific soundness.

Round 2

Reviewer 1 Report

I appreciate the changes brought by the authors in this short period of time. This confirms in my eyes their determination. The manuscript looks a lot better now. However, I would like to ask for a few small changes to fully informed the concerned reader and so that the conclusions will be fully supported by the results.

Firstly – please change the title to: “Evaluation of a Hypersensitivity Inhibitor Containing a Novel 3 Monomer that Induces Remineralization – a case series in pediatric patients” to fully inform about the type of the study.

Secondly, please divide the abstract dividing it into sections in accordance with the MDPI recommendations in the instructions for authors:

Abstract: The abstract should be a total of about 200 words maximum. The abstract should be a single paragraph and should follow the style of structured abstracts, but without headings: 1) Background: Place the question addressed in a broad context and highlight the purpose of the study; 2) Methods: Describe briefly the main methods or treatments applied. Include any relevant preregistration numbers, and species and strains of any animals used. 3) Results: Summarize the article's main findings; and 4) Conclusion: Indicate the main conclusions or interpretations. The abstract should be an objective representation of the article: it must not contain results which are not presented and substantiated in the main text and should not exaggerate the main conclusions.

In order to broaden the discussion and to improve the overall merit of the manuscript the study: “Long-Term Effectiveness of Treating Dentin Hypersensitivity with Bifluorid 10 and Futurabond U: A Split-Mouth Randomized Double-Blind Clinical Trial. J. Clin. Med. 2021, 10, 2085.” it is highly recommended for your attention.

Please change the first sentence of the conclusions to: “This novel hypersensitivity inhibitor with C-MET and MDCP suppressed hyperesthesia in young permanent teeth in presented cases.” Please add this sentence also to the abstract.

Please change the last sentence of the conclusions to: ”However, this report is based on a limited sample size, further evidence will be provided in an increasing number of homogenous cases in the future.”

Author Response

Reviewer #1

Firstly – please change the title to: “Evaluation of a Hypersensitivity Inhibitor Containing a Novel 3 Monomer that Induces Remineralization – a case series in pediatric patients” to fully inform about the type of the study.

Response to reviewer:

Thank you for your constructive suggestion. Title was modified according to your proposal.

Secondly, please divide the abstract dividing it into sections in accordance with the MDPI recommendations in the instructions for authors:

Response to reviewer:

Thank you for pointing it out. The abstract was divided according to the format of the MDPI.

In order to broaden the discussion and to improve the overall merit of the manuscript the study: “Long-Term Effectiveness of Treating Dentin Hypersensitivity with Bifluorid 10 and Futurabond U: A Split-Mouth Randomized Double-Blind Clinical Trial. J. Clin. Med. 2021, 10, 2085.” it is highly recommended for your attention.

Response to reviewer:

Thank you for introducing a paper that is very suitable for this manuscript. This information was added in the discussion paragraph and added to the references.

Please change the first sentence of the conclusions to: “This novel hypersensitivity inhibitor with C-MET and MDCP suppressed hyperesthesia in young permanent teeth in presented cases.” Please add this sentence also to the abstract.

Response to reviewer:

Thank you for suggestion. The first sentence of the conclusions was changed according to your instructions. The abstract was modified, so Keywords was also changed.

Please change the last sentence of the conclusions to: ”However, this report is based on a limited sample size, further evidence will be provided in an increasing number of homogenous cases in the future.”

Response to reviewer:

Thank you for your correction. According to your correction, the article was changed.

Reviewer 3 Report

As I said before this manuscript could be  only case report.

The authors agreed.

Best regards,

Author Response

Reviewer #3

As I said before this manuscript could be only case report.

The authors agreed.

Response to reviewer:

As you pointed out, this article is only a case report, so the manuscript type was changed to “case report”.